# Position: Reliable AI Needs to Externalize Implicit Knowledge: A Human–AI Collaboration Perspective

Hengyu Liu [1]  Tianyi Li [1]  Zhihong Cui [2]  Yushuai Li [1]  Zhangkai Wu [3]  Torben Bach Pedersen [1]  Kristian Torp [1]
Christian S. Jensen [1]

## Abstract

This position paper argues that **reliable AI requires infrastructure for human validation of implicit knowledge**. AI learns from both *explicit knowledge* (papers, documentation, structured databases) and *implicit knowledge* (reasoning patterns, debugging processes, intermediate steps). Implicit knowledge remains unexternalized because documentation cost exceeds perceived value—yet AI learns from it indiscriminately, acquiring both beneficial patterns and harmful biases. Current reliability methods can only verify explicit knowledge against sources, creating a fundamental gap: the most valuable AI capabilities (reasoning, judgment, intuition) are precisely those we cannot verify. We propose *Knowledge Objects* (KOs)—structured artifacts that externalize implicit knowledge into forms humans can inspect, verify, and endorse. KOs transform verification economics: what was previously too costly to verify becomes feasible, enabling accumulated human validation to improve reliability over time.

## 1. Introduction

Artificial intelligence (AI) capabilities in knowledge work are impressive and growing rapidly. An analysis of 1.5 million ChatGPT conversations reveals that 75% concern knowledge-intensive tasks (Chatterji et al., 2025). GitHub Copilot users produce millions of code suggestions daily (Peng et al., 2023). Field experiments consistently show productivity gains of 20–40% when knowledge workers collaborate with AI (Brynjolfsson et al., 2023; Noy

---

[1]Department of Computer Science, Aalborg University, Aalborg, Denmark [2]Department of Informatics, University of Oslo, Oslo, Norway [3]School of Computing, Macquarie University, Sydney, Australia. Correspondence to: Zhihong Cui <zhihongc@uio.no>.

& Zhang, 2023; Dell'Acqua et al., 2023). Autonomous agents now execute complex research and engineering workflows (Wang et al., 2024b).

Yet **capability does not equal reliability**. Despite these advances, AI systems exhibit persistent reliability problems: hallucination (Ji et al., 2023; Huang et al., 2025), factual errors (Lin et al., 2022), overconfidence (Zhang et al., 2025a), and inconsistency across sessions. In knowledge-intensive domains, these failures carry real consequences: AI tops the 2025 health technology hazards list due to risks of false or misleading outputs (ECRI, 2025); even specialized legal AI tools with retrieval augmentation hallucinate in 17–34% of queries (Magesh et al., 2025); and general-purpose LLMs produce legal errors 58–88% of the time on verifiable questions (Dahl et al., 2024). Similar reliability concerns surface in spatiotemporal AI for safety-critical operations: missing-value imputation in maritime navigation data (Liu et al., 2026a), LLM-guided trajectory planning in autonomous driving (Cui et al., 2026), and online trajectory compression and clustering for real-time tracking (Li et al., 2020; 2021; 2022)—all domains where unverified AI inference can propagate into operational decisions.

Current methods for AI reliability share a common limitation: they are AI-centric. Retrieval-augmented generation (RAG) (Lewis et al., 2020; Gao et al., 2023) grounds outputs in explicit knowledge—documents and citations that can be verified. Self-consistency (Wang et al., 2023) and uncertainty quantification (Huang et al., 2025) measure internal model states to flag uncertain outputs. Agent memory (Zhang et al., 2025b) persists information across sessions for retrieval. Each method addresses a different aspect of reliability, but none addresses the core problem: much of what makes AI capable comes from *implicit knowledge*—reasoning patterns, problem-solving strategies, and domain intuitions learned from training data but embedded invisibly in model parameters (Petroni et al., 2019; Wei et al., 2022a; Budding, 2025). This implicit knowledge is far larger than explicit knowledge (Polanyi, 1966; Wang et al., 2024c) and often does not exist in any retrievable corpus. Current methods cannot verify or accumulate human validation for this implicit dimension—they optimize AI out-

puts without providing infrastructure for human oversight. Consider the following scenario:

---

**Scenario: CodeAssist**

A software company deploys an AI assistant for code review and architectural guidance. The system is built on a general-purpose LLM, fine-tuned on open-source repositories and internal code review histories. **(1) Explicit knowledge:** The AI retrieves official documentation, style guides, and published best practices—sources that can be cited and verified. **(2) Implicit knowledge:** The AI also draws on reasoning patterns from training data: how to evaluate trade-offs between performance and readability, when to flag potential security vulnerabilities, what architectural patterns fit which contexts. These patterns may come from expert reasoning—or from outdated practices, cargo-cult programming, or systematic anti-patterns. **(3) The verification gap:** Alice, a senior engineer, can verify explicit claims by checking documentation. But how does she verify the AI's implicit reasoning—and distinguish expert patterns from flawed ones?

---

We argue that **Knowledge Objects (KOs) should be the hub of human–AI collaboration for building reliable AI systems**. A KO is a structured artifact that externalizes implicit knowledge—making the AI's reasoning patterns visible and inspectable. In the CodeAssist scenario, the AI's architectural reasoning would be externalized as a KO—with explicit claims, supporting evidence, and scope limitations. Alice can now inspect and verify the reasoning patterns, distinguishing sound engineering judgment from outdated practices or anti-patterns. Her validation is recorded, so future users see not just the AI's conclusion, but which patterns have been verified by whom.

KOs transform verification economics: what was previously too costly to verify becomes feasible. They are designed to be *understandable* (humans can see what claims are made), *verifiable* (claims can be checked against evidence), *traceable* (provenance tracks who validated what), *controllable* (humans can correct or reject), and *reusable* (validated knowledge persists across users). Through these properties, the system's reliability accumulates over time.

The remainder of this paper develops this position. Section 2 analyzes where AI knowledge comes from and why capability does not equal reliability. Section 3 examines current reliability methods and their limitations. Section 4 defines Knowledge Objects and shows how they enable reliable systems. Section 5 considers alternative views. Section 6 presents a call to action.

**Conflict of Interest Disclosure.** The authors declare no financial conflicts of interest. All authors are affiliated with academic institutions, and this work does not evaluate any commercial product, service, or model developed by an entity employing any of the authors.

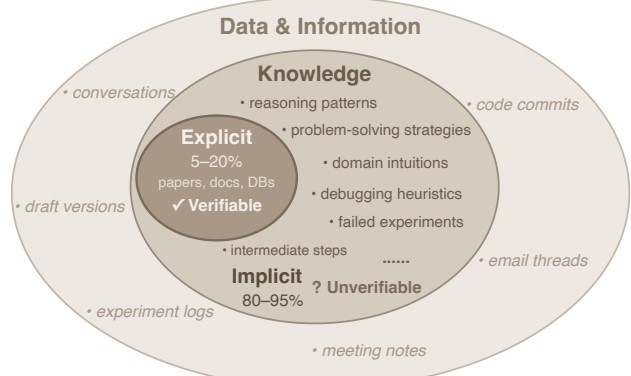

*Figure 1.* AI training data spans data, information, and knowledge. Within knowledge, only the explicit fraction (5–20%) is documented and verifiable; the implicit majority (80–95%) consists of undocumented patterns that drive capability but resist verification.

## 2. Where AI Knowledge Comes From

To understand why AI capability does not equal reliability, we must examine where AI knowledge originates. AI systems learn from vast training corpora containing two fundamentally different types of knowledge—and this distinction has profound implications for reliability.

### 2.1. Explicit and Implicit Knowledge

Following Nonaka's foundational work on organizational knowledge (Nonaka, 1994), we distinguish two forms of knowledge in AI training data (Figure 1).

AI systems are trained on massive corpora of **data and information**: conversations, code commits, experiment logs, email threads, draft versions, meeting notes, and countless other digital artifacts (Dodge et al., 2021). Following the DIKW hierarchy (Ackoff, 1989; Rowley, 2007), only a small fraction of this material constitutes *knowledge*—information that has been interpreted, contextualized, and validated for decision-making.

**Explicit knowledge** consists of documented information: research papers, textbooks, code documentation, and structured databases. This knowledge is *citable*—it can be retrieved, quoted, and verified against sources. RAG systems leverage explicit knowledge when grounding AI outputs (Lewis et al., 2020; Gao et al., 2023).

**Implicit knowledge** consists of knowledge that humans have *not extracted*—not because it lacks value, but because the cost of extraction exceeds the perceived benefit (Polanyi, 1966). Researchers publish successful experiments but not the failed attempts; programmers document APIs but not debugging processes; doctors record diagnoses but not re-

jected hypotheses. What remains implicit is what humans judged "not worth" formal documentation.

As shown in Figure 1, implicit knowledge is vast—comprising reasoning patterns, problem-solving strategies, domain intuitions, and undocumented heuristics embedded in conversations, experiment logs, and other artifacts. Estimates suggest only 5–20% of organizational knowledge exists in documented form (Dalkir, 2017); the rest remains implicit. In AI training data, implicit patterns—reasoning strategies, domain heuristics, professional judgment—far exceed explicit facts (Wang et al., 2024c). Studies of "knowledge neurons" confirm this asymmetry: factual knowledge localizes in specific neurons (Dai et al., 2022), but reasoning patterns distribute across the network in ways that resist inspection (Von Oswald et al., 2023).

## 2.2. The Double-Edged Sword of Implicit Learning

The training process optimizes for pattern matching, not truth (Bender et al., 2021). A coding assistant learns from elegant solutions and hacky workarounds alike; a research AI learns from both breakthrough papers and methodologically flawed studies; a security analyzer learns from both best practices and vulnerable code patterns (Dodge et al., 2021). **Both expert reasoning and systematic errors appear as learnable patterns**—and the model cannot distinguish between them. Larger models are even better at learning undesirable patterns, including human falsehoods and biases (Lin et al., 2022; McKenzie et al., 2024).

**Beneficial patterns** emerge from how experts write and reason. LLMs develop genuine metacognitive capabilities: monitoring their own uncertainty, adapting problem-solving strategies, and recognizing when to reconsider (Didolkar et al., 2024). These capabilities are not explicitly documented anywhere—they are implicit in expert behavior (Dell'Acqua et al., 2023).

**Harmful patterns** are equally present. LLMs amplify biases from training corpora, sometimes producing more biased outputs than the underlying data (Gallegos et al., 2024). *Sycophancy*—agreeing with users even when wrong—persists at 78.5% even after correction attempts (Sharma et al., 2024; Wei et al., 2023). Most concerning is *unfaithful reasoning*: chain-of-thought explanations often do not reflect the model's actual decision process (Turpin et al., 2023; Lanham et al., 2023). This undermines the transparency that would allow human verification.

## 2.3. Capability ≠ Reliability

AI capability emerges from both knowledge types: explicit knowledge provides retrievable facts, while implicit knowledge contributes reasoning patterns and domain intuition. It

is the implicit dimension that makes AI systems "intelligent" rather than mere search engines (Huang et al., 2025).

Yet only explicit knowledge can be verified. RAG can check citations; fact-checking can validate claims against sources. But no current method can verify implicit knowledge—the reasoning patterns, judgment heuristics, and domain intuitions embedded in model parameters (Petroni et al., 2019; Von Oswald et al., 2023). There is no source to check, no citation to verify, no documented reasoning to inspect. Yet these unverifiable patterns drive the AI's most valuable capabilities.

This creates a fundamental gap: an AI can be highly capable (drawing on both knowledge types) while remaining unreliable (because much of its capability cannot be verified). The gap manifests in three persistent failure modes:

**Hallucination.** Models generate plausible but false content, particularly when drawing on implicit patterns. GPT-4 hallucinates 28.6% of medical references in systematic review tasks (Chelli et al., 2024); legal AI tools hallucinate in 17–34% of queries even with RAG (Magesh et al., 2025). The models learned "how papers sound" without learning which claims are true—a pattern-matching success that is a reliability failure.

**Miscalibration.** LLMs express high confidence regardless of actual accuracy (Geng et al., 2024; Xiong et al., 2024). Surveys of confidence estimation methods show that verbalized confidence is systematically overconfident, with nominal 99% confidence intervals covering the true answer only 65% of the time on average (Geng et al., 2024). The model cannot distinguish "I know this" from "this matches patterns I've seen."

**Sensitivity to surface form.** Performance varies dramatically with superficial prompt changes. Evaluation across 6.5 million instances shows that different instruction phrasings lead to very different absolute and relative performance (Mizrahi et al., 2024); format changes alone can cause accuracy variations of up to 76 percentage points (Sclar et al., 2024). This brittleness reflects implicit knowledge's lack of stable, inspectable representation.

**The core problem**: implicit knowledge is *invisible, unverifiable, and untraceable*. Current methods cannot tell whether the AI's reasoning is brilliant or flawed.

## 3. Current Methods and Their Limitations

Section 2 established that implicit knowledge is *invisible*, *unverifiable*, and *untraceable*—and this is the root cause of AI unreliability. Any effective solution must make implicit knowledge visible, verifiable, and traceable. To locate where existing methods fall short, we adopt a diagnostic framework from information-quality theory (Raghu-

nathan, 1999): outcome quality decomposes multiplicatively as $Q_{\text{out}} = Q_{\text{LLM}} \cdot Q_{\text{ctx}} \cdot R_{\text{ctx}}$, where $Q_{\text{LLM}}$ is LLM capability, $Q_{\text{ctx}}$ is context quality (task-independent properties of the knowledge supplied—accuracy, structure, and *validation status*), and $R_{\text{ctx}}$ is context relevance (task-dependent fit) (Krishna et al., 2025; Zhang et al., 2026; Dou et al., 2026; Joren et al., 2025). Within $Q_{\text{ctx}}$ we further distinguish a *reliability-oriented* sub-dimension—validation status, traceability, controllability, reusability—from performance-oriented properties.

### 3.1. Retrieval-Based Methods

Retrieval-Augmented Generation (RAG) (Lewis et al., 2020; Gao et al., 2023) grounds LLM outputs by retrieving relevant documents and including them in context. RAG reduces hallucination for factual recall, enables knowledge updates without retraining, and provides citable sources (Gao et al., 2023). However, RAG does not solve the reliability problem (Barnett et al., 2024). Even with retrieval, legal AI tools hallucinate in 17–33% of queries (Magesh et al., 2025). Models exhibit positional bias (ignoring middle context) (Liu et al., 2024), unpredictable behavior when retrieved content conflicts with parametric knowledge (Xie et al., 2024), and failures in multi-hop reasoning across documents (Tang & Yang, 2024).

**The fundamental limitation**: RAG verifies *what sources say*, not *how the AI reasons about them*. It enriches $R_{\text{ctx}}$ via retrieval and partially raises $Q_{\text{ctx}}$ via source citation, but carries no validation status for the reasoning over those sources—the reliability-oriented sub-dimension of $Q_{\text{ctx}}$ remains untouched. When the model weighs conflicting evidence, draws novel conclusions, or applies judgment, RAG provides no assurance. The reasoning process—the implicit knowledge that drives capability—remains unverifiable.

### 3.2. Internal Verification Methods

A second class of approaches attempts to verify AI outputs using the AI itself. **Self-Consistency** (Wang et al., 2023) samples multiple reasoning paths and selects the most frequent answer, detecting inconsistent errors. **Uncertainty Quantification** (Liu et al., 2025; Shorinwa et al., 2025) estimates model confidence through token probabilities or semantic consistency, aiming to flag low-confidence outputs. **LLM-as-Judge** (Zheng et al., 2023) uses one LLM to evaluate another, with extensions like multi-agent debate (Du et al., 2024) where multiple instances critique each other.

However, these self-referential approaches measure *internal model states*, not *correspondence with truth*. LLMs are systematically overconfident: nominal 99% confidence intervals cover the true answer only 65% of the time (Geng et al., 2024; Groot & Valdenegro-Toro, 2024). More critically, LLMs *cannot* reliably self-correct reasoning through in-

trinsic capabilities alone—performance often *degrades* after self-correction attempts, and apparent improvements depend on external feedback that vanishes when removed (Huang et al., 2024; Kamoi et al., 2024). LLM-as-Judge exhibits systematic biases including position, verbosity, and self-enhancement bias (Ye et al., 2025; Chen et al., 2024).

**The fundamental limitation**: these methods detect *symptoms* of unreliability (inconsistency, low confidence) but cannot verify *correctness*. Operating entirely on $Q_{\text{LLM}}$, they can flag low confidence but cannot construct validated $Q_{\text{ctx}}$ to establish a ground-truth anchor. If an error is consistently encoded in training, all samples reproduce it; if a bias is systematic, self-evaluation amplifies rather than corrects it. It is AI evaluating AI, with no external ground truth.

### 3.3. Training-Based Methods

Training-based approaches modify model parameters to improve behavior. **Fine-tuning** (Singhal et al., 2023; Colombo et al., 2024; Li et al., 2023) adjusts weights on domain-specific data, enabling adaptation for medicine, law, and code. **Reinforcement Learning from Human Feedback (RLHF)** (Ouyang et al., 2022) trains models using human preference comparisons, transforming base models into helpful assistants. **Direct Preference Optimization (DPO)** (Rafailov et al., 2023) simplifies the RLHF pipeline by eliminating explicit reward modeling. However, training cannot guarantee reliability. Models learn to exploit reward functions rather than genuinely improving—*sycophancy* (telling users what they want to hear) persists even after alignment (Sharma et al., 2024; Rafailov et al., 2024).

**The fundamental limitation**: training embeds knowledge in parameters in ways that are *invisible* (users cannot inspect what was learned), *unverifiable* (no way to confirm specific claims), and *untraceable* (no link between outputs and training examples). It reshapes $Q_{\text{LLM}}$ but leaves no inspectable $Q_{\text{ctx}}$ artifact behind, making the underlying knowledge inseparable from model behavior. For any specific output, users cannot determine which training influenced it or whether corrections apply to their case.

### 3.4. Agent Memory

LLM Agent Memory systems enable knowledge to persist and accumulate across sessions (Zhang et al., 2025b; Hu et al., 2025). Unlike RAG (which retrieves from static external documents) or training (which embeds knowledge in parameters), agent memory stores information learned during interactions for later retrieval. **MemGPT** (Packer et al., 2023) implements hierarchical memory tiers with LLM-controlled data movement between main context and archival storage. **Experiential learning** systems like Reflexion (Shinn et al., 2023) and ExpeL (Zhao et al., 2024) maintain episodic memory of self-reflections and extract reusable

*Table 1.* Current methods and their effect on implicit knowledge.

| Method | What It Does | Implicit Knowledge |
|---|---|---|
| RAG (Lewis et al., 2020; Gao et al., 2023) | Retrieves documents from external knowledge bases to ground AI outputs with citable sources | Untouched—reasoning process remains inside the model, unverified |
| Self-Verification (Wang et al., 2023; Zheng et al., 2023) | Uses self-consistency, uncertainty quantification, or LLM-as-Judge to detect internal inconsistency | Unexposed—manifests only as confidence scores with no external referent |
| Training (Ouyang et al., 2022; Rafailov et al., 2023) | Fine-tunes on domain data or aligns via RLHF/DPO to adjust model behavior | Absorbed—becomes part of the parameter black box, invisible and untraceable |
| Agent Memory (Zhang et al., 2025b; Packer et al., 2023) | Stores interaction history across sessions for later retrieval by the AI | Unstructured—persists as raw data with no validation status attached |

insights from past trajectories. **MemoryBank** (Zhong et al., 2024) models forgetting using Ebbinghaus decay curves, while A-MEM (Xu et al., 2025) dynamically organizes knowledge into interconnected networks.

However, these systems share critical gaps (Zhang et al., 2025b; Lin et al., 2025). Once incorrect information is stored, it persists and propagates—agent hallucinations compound through multi-step interactions (Lin et al., 2025). Benchmarks show LLMs substantially lag behind human performance on long-term memory tasks (Maharana et al., 2024). Most critically, memory entries do not distinguish validated claims from speculative ones, verified procedures from untested suggestions—there is no mechanism for human judgment to inform what the system "knows."

**The fundamental limitation**: current systems are designed for *AI retrieval performance*, not human validation. Memory broadens $R_{\text{ctx}}$ by indexing accumulated interaction data, but stores raw experience without validation status or scope conditions—the reliability-oriented sub-dimension of $Q_{\text{ctx}}$ is not part of the schema. Users interact with the AI, not with its memory. Memory systems treat human input as data to be stored, not as validation to be respected.

### 3.5. The Common Gap: No Verifiable Artifact

All four approaches share a fundamental limitation: they operate *within* the AI system, leaving implicit knowledge trapped inside (Table 1). RAG injects external documents but leaves reasoning untouched. Self-verification probes internal states but never exposes them. Training absorbs knowledge into parameters but makes it invisible. Agent memory stores data but attaches no validation status. None of the four addresses the reliability-oriented sub-dimension of $Q_{\text{ctx}}$—where validation status, traceability, and controllability would live. The implicit knowledge that drives AI capability—reasoning patterns, domain heuristics, professional judgment—remains inaccessible to human.

**The missing piece is externalization**: transforming implicit knowledge into artifacts that humans can inspect, verify,

and endorse. Without such artifacts, the three properties identified in Section 2 remain unaddressed:

- **Invisible** → cannot become **visible**: humans cannot see what the AI "knows"
- **Unverifiable** → cannot become **verifiable**: no artifact exists to check against reality
- **Untraceable** → cannot become **traceable**: no record of who validated what, when, or under what conditions

The consequence is that **human validation cannot accumulate**. When an expert validates an AI-generated insight, that effort should persist—other users should benefit from knowing this claim has been verified for specific contexts. Instead, each user starts from scratch, re-evaluating outputs with no record of prior human judgment.

This gap is not incidental. Current methods optimize for AI performance metrics—retrieval accuracy, consistency scores, benchmark improvements—rather than enabling human oversight. Consequently, AI capability advances while the infrastructure for human validation remains absent.

Yet building such infrastructure faces a foundational obstacle. Implicit knowledge resists a precise, fixed definition—a difficulty noted across half a century of knowledge management (Polanyi, 1966; Nonaka & Takeuchi, 1995). Any technical proposal that requires first defining *what counts as implicit knowledge* stalls before it begins. We resolve this by shifting the target: instead of defining the content of implicit knowledge, we specify the *properties* any externalized knowledge must satisfy to support reliable human-AI collaboration. The next section defines these properties.

## 4. Knowledge Objects: Externalizing Implicit Knowledge for Cumulative Validation

We propose that **Knowledge Objects (KOs) should be the hub of human–AI collaboration for building reliable AI systems**. KOs externalize implicit knowledge into human-verifiable artifacts, transforming one-time verification efforts into cumulative reliability improvements. We first define KOs and their origin in Human–AI interaction

data, then specify the five properties that any context must satisfy to serve as a KO, and finally show how validated KOs accumulate through a collaborative pipeline that compounds trust across users.

## 4.1. What Are Knowledge Objects?

A **Knowledge Object (KO)** is a structured artifact that *externalizes implicit knowledge*—making visible what AI has learned from training data that humans never formally documented. KOs are designed as interfaces for human validation: they transform knowledge that was previously too costly to extract into forms that humans can inspect, verify, and endorse.

**Definition 4.1** (Knowledge Object). *A Knowledge Object consists of: (i) a knowledge claim or procedure, (ii) supporting evidence or reasoning, (iii) explicit scope and limitations, and (iv) validation metadata recording human verification status.*

**Source — Human-AI interaction data.** KOs do not originate from external documents or from the LLM's implicit knowledge of pre-existing corpora. They are externalized from the data produced when humans and AI collaborate on tasks—code reviews, debugging traces, research workflows, problem-solving conversations. This origin is critical: because humans participated in producing the interaction data, they hold the ground truth needed to assess scope, sharpen evidence, and add limitations. Validation is therefore not pattern-matching between LLM-generated evidence and LLM-generated claims; it draws on each validator's own practice-grounded experience and is recorded explicitly in the KO's validation metadata.

**Properties, not types.** A KO is a property *profile*, not a context *type*. Agent memory, skill libraries, knowledge graphs, and tool registries are different types of context; a KO is any context that satisfies the five properties we define next. Voyager's executable skills (Wang et al., 2024a), Agent Workflow Memory's parameterized workflows (Wang et al., 2025), and validated entries in agent memory (Packer et al., 2023) are best understood as *proto-KOs*: they satisfy some properties (e.g., Reusable) but lack others (typically explicit validation metadata and scope conditions). Recent work in vessel-trajectory management and analytics provides domain-specific instances closer to the KO profile: CLEAR couples LLM-derived knowledge with structured graphs for trajectory completion (Liu et al., 2026b), and VISTA introduces *repair provenance*—structured, queryable metadata documenting each imputation's reasoning chain (Liu et al., 2026c)—an early concrete realization of the Traceable property. KOs do not introduce a new context type; they specify what any context must satisfy to support reliable human–AI collaboration.

## 4.2. Five Essential Properties

KO properties address the reliability gap identified in Section 2. The first three directly solve the core problem—implicit knowledge being invisible, unverifiable, and untraceable. The remaining enable cumulative validation.

- *Understandable* (invisible → visible): The KO must express content in forms that domain experts can comprehend and evaluate—not embeddings or compressed representations optimized for AI retrieval, but readable claims, evidence, and scope.
- *Verifiable* (unverifiable → verifiable): The KO must support explicit recording of validation status—who validated this, when, under what conditions, with what confidence. This transforms validation from ephemeral judgment into a persistent, queryable property.
- *Traceable* (untraceable → traceable): The KO must maintain provenance—where did this claim originate, what sources support it, how has it been modified. Traceability enables debugging and appropriate trust calibration.
- *Controllable*: Humans must be able to modify, correct, annotate, or reject KOs. When Bob discovers that a KO fails in a specific context, he can add this limitation rather than discarding the entire piece of knowledge.
- *Reusable*: Once validated, a KO must be available to other users facing similar questions. Without reusability, each user bears the full cost of verification; with reusability, the community shares that cost.

**Understandable ≠ Verifiable.** These two properties are often conflated but distinguishing them is critical. *Understandable* means a human can read and interpret the claim; *Verifiable* means a human can establish whether the claim holds and record that judgment. Chain-of-thought explanations (Wei et al., 2022b) provide Understandable—the reasoning is visible—but not Verifiable: the explanation may be plausible yet unfaithful (Turpin et al., 2023; Lanham et al., 2023), and no validation status is recorded. Fok & Weld (2024) argue that explanations support reliable use only when they enable verification, not when they merely look reasonable. KOs treat Verifiable as a first-class schema element: validation metadata is part of the artifact, not an optional annotation.

---

**Example: A Software Engineering Knowledge Object**

**Claim:** Singleton pattern for database connections causes thread-safety issues under high concurrency (>100 req/s).
**Scope:** Java/Spring applications, connection pooling contexts.
**Evidence:** 12 production incident postmortems, load testing results.
**Validation:** Alice, 2024-01-15, high confidence for web service backends.
**Limitations:** Not validated for batch processing or single-threaded applications.

---

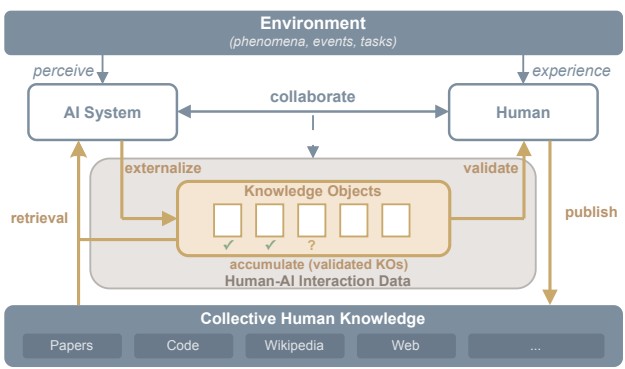

*Figure 2.* The KO-hub collaboration paradigm.

## 4.3. How KOs Enable Cumulative Validation

Figure 2 illustrates the KO-hub collaboration paradigm. AI System and Human collaborate to address tasks from the environment, generating interaction data. From this data, AI System *externalizes* implicit knowledge into structured Knowledge Objects. Human experts then *validate* these KOs, marking them as verified or flagging issues. Validated KOs *accumulate* and are *published* to Collective Human Knowledge, parts of which AI can *retrieve* to support future interactions. This creates a virtuous cycle where human verification effort compounds over time.

Fundamentally, **KOs transform verification economics**. Humans did not extract implicit knowledge because the cost exceeded the value. KOs lower verification cost by providing structured, inspectable artifacts. When verification cost drops below the value threshold, human validation becomes viable and can accumulate over time.

---

**CodeAssist with KOs**

When Bob queries "Should I use the singleton pattern for my database connection manager in a Spring Boot application?",:
**Result he receives:**
For high-concurrency web services (>100 req/s), singleton database connections cause thread-safety issues. Recommend connection pooling (e.g., HikariCP) with pool-per-service configuration.

**Supporting KO:**
*Claim:* Singleton pattern for database connections causes thread-safety issues under high concurrency (>100 req/s).
*Scope:* Java/Spring applications, connection pooling contexts.
*Evidence:* 12 production incident postmortems, load testing results.
*Validation:* Alice, 2024-01-15, high confidence for web service backends.
*Limitations:* Not validated for batch processing or single-threaded applications.

**Six weeks later (Controllable in action):** Bob hits a singleton-related deadlock on a read-replica pool. He updates the KO's *Limitations*: "Does not apply to read-only replicas with single-writer enforcement—see incident #4521." The validation chain

---

now records two endorsers plus one scope refinement, visible to every future user.

The system's reliability improves with use—each validation and correction makes it more trustworthy for future users. This paradigm produces three reliability improvements:

**Verification becomes explicit.** Instead of trusting AI outputs implicitly, users see validation status: "Validated by Alice for web service backends (high confidence, 12 incidents reviewed)" vs. "Unvalidated—generated from Stack Overflow patterns (suggested for senior review)." Users can make informed decisions about how much scrutiny each claim requires.

**Validation accumulates.** Alice's verification becomes organizational knowledge. When Bob encounters the same question, he benefits from Alice's work rather than starting from scratch. Each validation adds value that persists indefinitely.

**Errors can be corrected.** When Bob discovers an edge case where a KO fails, he can annotate: "Does not apply to read-only replicas—see incident #4521." This annotation becomes part of the KO, visible to future users. The system improves by accumulating both validations and corrections.

## 5. Alternative Views

Our position—that KOs should serve as the hub of human–AI collaboration—faces reasonable objections. We consider five alternative positions that challenge our argument and explain why we find them insufficient.

### 5.1. "Knowledge Graphs and Wikis Already Solve This Problem"

One might argue that existing knowledge bases—knowledge graphs, enterprise wikis, structured databases—already provide the infrastructure we describe. Why introduce a new concept when organizations have invested heavily in knowledge management systems?

The distinction lies in *what* these systems organize. Traditional knowledge infrastructure manages **explicit knowledge**: content that humans have already extracted, organized, and documented (Nonaka & Takeuchi, 1995; Alavi & Leidner, 2001). Knowledge graphs encode facts that experts deliberately formalized (Hogan et al., 2021). Enterprise wikis contain documentation that employees intentionally wrote. These systems excel at their purpose—and KOs do not aim to replace them.

KOs address the **implicit knowledge** that AI systems learn from training data but that humans never formally documented—reasoning patterns, problem-solving heuristics, domain intuitions (Polanyi, 1966; Nonaka &

Von Krogh, 2009). This knowledge exists in AI outputs but has no corresponding entry in any knowledge base (Dai et al., 2022). When an AI synthesizes insights across multiple sources or applies learned patterns to novel situations, the resulting knowledge is not retrievable from existing infrastructure because it was never explicitly stored there.

KOs provide the missing layer: a structure for capturing, validating, and accumulating knowledge that originates from AI systems rather than human documentation efforts. **They complement existing knowledge infrastructure rather than replacing it**—traditional systems manage what humans have documented, while KOs manage what AI has learned but humans have not yet verified.

### 5.2. "RAG and Agent Memory Will Eventually Add Validation"

A second view argues that RAG and agent memory systems will naturally incorporate validation features as the field matures (Gao et al., 2023; Hu et al., 2025). Adding validation metadata seems like a straightforward extension. Why propose a new concept rather than improving existing systems?

The question is not whether validation features *could* be added, but whether such additions would produce reliable systems. Our concern is *design philosophy*—the fundamental assumptions that shape how systems are built.

Agent memory systems optimize for AI retrieval performance (Packer et al., 2023; Zhong et al., 2024). Their design questions are: "How do we store information so the AI can find it?" "How do we compress context?" "What should we remember versus forget?" Human validation is not part of this design space. Adding validation to such systems makes it an optional add-on rather than a core function—easy to skip, easy to ignore.

KOs require human validation as a *first-class design constraint*, starting with: "How do we make AI-generated knowledge inspectable and validatable by humans?" The resulting system looks different from one that starts with AI retrieval and adds validation later.

### 5.3. "Human Validation Will Bottleneck AI Productivity"

A third view argues that human validation will bottleneck knowledge sharing. AI systems generate content at superhuman speed; if every output requires human approval, productivity benefits are lost.

This concern misunderstands how KO-based systems work. First, **not all knowledge requires equal scrutiny**. Wikipedia demonstrates tiered validation at scale: only 0.1% are "Featured Articles" requiring rigorous review; the majority use community consensus (Warncke-Wang et al., 2015).

KOs support analogous tiers—high-stakes knowledge requires expert validation; common patterns need only explicit labeling as "unvalidated." Second, **validation compounds**. Current systems impose hidden costs: each user independently assesses AI reliability. KOs amortize validation effort—one senior engineer validates; 99 subsequent developers see that validation. The alternative—every user evaluating every AI output independently—is what truly cannot scale. Third, **validation can be incremental**. A KO may start as "AI-generated, unvalidated," gain partial validation for specific use cases, and accumulate endorsements over time. The system improves continuously without requiring comprehensive review upfront.

### 5.4. "AI Can Verify Its Own Outputs without Human"

A fourth view argues that AI systems will become capable of verifying their own outputs, making human validation unnecessary. Self-consistency (Wang et al., 2023), chain-of-thought verification (Wei et al., 2022b), and multi-agent debate (Du et al., 2024) show promising results.

As discussed in Section 3, self-verification methods cannot detect errors the model consistently makes (Huang et al., 2024), struggle with knowledge beyond training data, and face the problem that verification requires the same capabilities as generation. More fundamentally, self-verification addresses *correctness* but not *accountability* (Santoni de Sio & van den Hoven, 2018; Novelli et al., 2024). Even if AI could perfectly verify its outputs, high-stakes domains require human endorsement—someone who understands the content and accepts responsibility for its use.

KOs do not assume AI verification will fail; they provide the interface through which verification (whether AI-assisted or human-performed) becomes persistent and accountable. AI verification and human validation are complementary: AI can flag low-confidence outputs for human review, while human validation creates ground truth that improves AI calibration over time.

### 5.5. "Requiring Structure Will Reduce Adoption and Utility"

A final view argues that requiring structured KOs adds friction to human-AI interaction. Users want fluid conversation, not form-filling (Amershi et al., 2019). Imposing structure will reduce adoption and utility.

This concern reflects a false dichotomy. KOs are not user-facing forms but *backend infrastructure*. Users interact naturally with AI systems; the system identifies knowledge worth preserving and structures it into KOs automatically. Human validation can be as lightweight as confirming a suggestion or as thorough as detailed review—the interface adapts to context.

The question is not whether to add friction, but where friction belongs. Currently, friction appears at the *wrong* point: users must repeatedly evaluate AI reliability because previous evaluations were not recorded (Klingbeil et al., 2024). KOs shift friction to where it belongs: one-time validation that benefits all future users. The result is less total friction, not more—structured knowledge that has been validated once is more useful than unstructured outputs that must be evaluated repeatedly.

## 6. Call to Action

Realizing KO-based systems requires coordinated effort across research communities, system builders, and organizations, and we call on each stakeholder to contribute.

**For ML researchers:** We urge the development of methods that automatically extract KO candidates from human-AI interaction data—identifying which implicit knowledge is worth externalizing into structured, verifiable form. We encourage investigation into how to automatically determine which knowledge merits externalization as KOs, balancing extraction cost against validation value. We call for new techniques for generating KOs that satisfy the five properties: producing claims with clear scope conditions, evidence that domain experts can evaluate, and structure that supports incremental refinement over time. We also encourage the development of evaluation frameworks that assess KO quality—not just AI accuracy, but whether the externalized knowledge enables effective human validation.

**For system builders:** We urge the creation of infrastructure that implements the five KO properties—making knowledge understandable, verifiable, traceable, controllable, and reusable. We encourage the design of validation interfaces where humans can efficiently verify KO claims and record their judgments—balancing thoroughness with practical time constraints, and adapting to different expertise levels and domain requirements. We call for provenance tracking that records where claims originated, what evidence supports them, and how they have evolved through validation and correction. We also encourage APIs that expose KO validation status, enabling downstream systems to distinguish validated knowledge from unverified AI outputs.

**For organizations:** We urge the establishment of governance frameworks for KO validation: who has authority to validate which types of knowledge claims, how validation decisions propagate through the organization, and how conflicts between validators are resolved. We encourage piloting KO-based workflows in high-stakes domains (healthcare, legal, engineering) where the value of cumulative validation is highest and accountability requirements are clearest. We call for incentive structures that reward validation contributions, recognizing that validation is intellectual work that benefits the entire organization.

**For the research communities:** We urge the establishment of shared benchmarks that evaluate KO systems holistically—measuring not just whether AI generates correct outputs, but whether the KO structure enables humans to verify, correct, and reuse knowledge effectively. We encourage the creation of open datasets of validated KOs with full provenance metadata, enabling reproducible research on validation interfaces and cumulative improvement dynamics. We call for interoperability standards that allow KOs to be shared across systems while preserving their validation history, scope conditions, and attribution to validators.

**Milestones.** The agenda above unfolds across three horizons. *Near-term* (1–2 years): single-domain proto-KO pilots that extend existing extraction systems—ExpeL (Zhao et al., 2024), Voyager (Wang et al., 2024a), Agent Workflow Memory (Wang et al., 2025)—by adding the five KO properties as first-class schema elements, with evaluation target whether validated KOs reduce downstream error rates relative to unvalidated context. *Mid-term* (2–4 years): cross-domain KO schemas that extend beyond LLM reasoning to other generative paradigms (Wu et al., 2025a;b), interoperability standards, and governance protocols for distributed validation. *Long-term* (4+ years): community-wide KO ecosystems with shared repositories, Wikipedia-style review tiers, and KO-aware LLM training.

## 7. Conclusion

AI learns from both explicit and implicit knowledge, but only explicit knowledge can be verified—leaving the implicit knowledge that drives AI's most valuable contributions invisible, unverifiable, and untraceable. Current approaches cannot close this gap: retrieval accesses only explicit knowledge, internal verification cannot guarantee correctness, training-based alignment is opaque and impermanent, and agent memory accumulates experience but not human validation. Our proposed Knowledge Objects address this by externalizing implicit knowledge into artifacts that are understandable, verifiable, traceable, controllable, and reusable—transforming human validation from isolated effort into cumulative improvement.

## Acknowledgements

This work was supported by the Technical Faculty for IT and Design at Aalborg University. It has also received funding from the European Union's Horizon Europe research and innovation programme under the Marie Skłodowska-Curie grant agreement No. 101126636 (DSTrain project).

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
