# OpenReview forum: "Position: Reliable AI Needs to Externalize Implicit Knowledge: A Human–AI Collaboration Perspective"
_ICML.cc/2026/Position_Paper_Track — ICML 2026 Position Paper Track regular_

### Official Review · Reviewer_DVeA · 2026-03-12

**Significance:** 2
**Argument Clarity:** 2
**Rating:** 4
**Confidence:** 3

**Questions:**

See Weaknesses above.

**Alternative Views Section:**

Yes

**Compliance With Llm Reviewing Policy A Conservative:**

Affirmed.

**Discussion Potential:**

3

**Final Justification:**

Some questions remain—specifically, whether the five properties of KOs can indeed effectively transform knowledge from implicit to explicit. This is a fundamental claim that requires ongoing validation through future research and practical implementation.

As such, I maintain my positive score.

**Paper Summary:**

This paper argues that AI reliability is limited by the inability to verify implicit knowledge, which underlies reasoning, judgment, and intuition. It proposes Knowledge Objects (KOs)—structured artifacts that externalize implicit knowledge for human inspection and validation, making verification feasible and improving AI reliability over time.

**Position:**

Yes

**Position In Title:**

Yes

**Related Work:**

3

**Strengths And Weaknesses:**

Strengths
1.The authors cite numerous existing studies to support their arguments.
2.The authors provide a detailed categorical description.

Weaknesses
1.The definition of implicit knowledge is too broad and not fixed. For example, in code debugging, errors can occur in many ways, and each person may make different mistakes, making implicit knowledge difficult to control.
2.The four methods summarized in Section 3 cannot fundamentally be considered learning and verification of implicit knowledge; they still primarily deal with explicit knowledge.
3.The logical flow throughout the paper is somewhat difficult to follow.
4.Inclusion of more concrete case studies would improve the work.
5.The externalization of implicit knowledge lacks sufficient validation and reliable implementation pathways.

**Support:**

3

---

> ### Author Rebuttal · Authors · 2026-03-31
>
> We thank the reviewer for the constructive feedback.
>
> **W1.** Implicit knowledge, like knowledge itself, resists precise and fixed definition — this has been a philosophical challenge throughout history (Polanyi 1966 [1]: "we can know more than we can tell"; Alavi & Leidner 2001 [2]: "there is no universal definition of knowledge"). To achieve operability, our paper **focuses directly on five properties of KOs**: when a piece of information satisfies all five properties, it naturally becomes explicit knowledge that humans can trust — **KOs offer a bridge from implicit to explicit**.
>
> The reviewer's code debugging example illustrates this well. Alice discovers a singleton thread-safety bug in her Java/Spring project under high concurrency — without KO, this remains personal implicit knowledge, uncontrollable and invisible to others. With a KO, the system externalizes it with explicit properties: **Understandable** ("singleton causes thread-safety issues under high concurrency"), **Verifiable** (Alice records: "verified in production, Java/Spring, >100 req/s"), **Traceable** (originated from Alice's code review session), **Controllable** (when Bob finds it does not apply to read-only replicas, he adds this limitation rather than discarding the KO), and **Reusable** (team members facing similar concurrency issues can find and build on this validated KO). Each person's errors differ, but the five properties provide a uniform governance structure that makes diverse implicit knowledge controllable across scopes — from personal accumulation to community sharing.
>
> **W2.** As stated in W1, implicit knowledge cannot be defined precisely. Yet proposing a technical solution requires a precisely defined target — therefore no existing method can directly "solve" implicit knowledge. **This is the dilemma the community faces.** However, effectively leveraging implicit knowledge is immensely valuable, especially in the AI era. We therefore propose the KO concept, focusing on five properties. As stated in W1, KOs bridge implicit and explicit knowledge through its five properties. KOs provide an **anchor point** for the ML/AI community to leverage implicit knowledge and achieve Reliable AI.
>
> **W3 and W4.** We appreciate the reviewer pointing out these issues. For W3 (logical flow), as our responses to W1 and W2 demonstrate, key arguments — such as "implicit knowledge cannot be precisely defined", "proposing a technical solution requires a precisely defined target — therefore no existing method can directly solve implicit knowledge", and "To achieve operability, KOs focuse on five properties" — were not explicitly articulated.
>
> In the revised version, we will restructure the argumentation to make these threads explicit, particularly the transitions from Section 2's implicit knowledge problem to Section 3's method summary, and from Section 3.5's common gap to Section 4's KO proposal. For W4 (case studies), the Alice/Bob example is used through out the paper. We will enrich it to more clearly illustrate KO construction and maintenance.
>
> **W5.** This direction is indeed in its early stages — which is why we, in a position paper, call on the community to explore it. However, it does not start from zero — existing systems already show early signs: see Voyager's [3] self-verified skill library and ExpeL's [4] community-voted rules. The five properties of KOs provide an evolution direction for these early efforts — progressively adding governance properties from proto-KO toward full-KO compliance. For concrete milestones, please see the response to Reviewer mZZH W1.
>
> **References:**
>
> [1] *The Tacit Dimension*. Univ. of Chicago Press, 1966.
>
> [2] Knowledge management and knowledge management systems. *MIS Q.* 2001.
>
> [3] Voyager: An open-ended embodied agent with LLMs. *Trans. Mach. Learn. Res.* 2024.
>
> [4] ExpeL: LLM agents are experiential learners. *AAAI* 2024.

---

> > ### Author Rebuttal · Reviewer_DVeA · 2026-04-03
> >
> > Thanks for the detailed and thoughtful responses.
> >
> > The majority of our concerns have been satisfactorily addressed. The clarifications regarding the philosophical nature of tacit knowledge and the operational role of the five KO properties are particularly helpful in understanding the paper's positioning.
> >
> > However, some questions remain—specifically, whether the five properties of KOs can indeed effectively transform knowledge from implicit to explicit. This is a fundamental claim that requires ongoing validation through future research and practical implementation.
> >
> > As such, I maintain my positive score.

---

### Official Review · Reviewer_Qz8u · 2026-03-12

**Significance:** 4
**Argument Clarity:** 4
**Rating:** 4
**Confidence:** 4

**Questions:**

See weaknesses above. I'll be glad to increase my score if addressed.

**Alternative Views Section:**

Yes

**Compliance With Llm Reviewing Policy A Conservative:**

Affirmed.

**Discussion Potential:**

4

**Final Justification:**

I believe the authors addressed my concern regarding the scope of the paper. I believe, when the scope is precisely defined, the paper is an effective position paper, so I am changing my score.

**Paper Summary:**

The authors present the position that a reliable AI system needs infrastructure for human validation of implicit knowledge. They look at AI systems which interact with external users, and propose that in order that its reliability increases, there needs to be a verification mechanism on the backend of the system, where the AI produces Knowledge Objects (KOs) which are structured documents representing internal knowledge that was used when completing a task. These KOs can be validated by a human, and then can be retrieved by the LLM to perform subsequent requests  while adhering to the KO. The authors contrast KOs with other AI system mechanisms, and discusses the cascading effect of validating individual KOs for a generally more trustworthy system.

**Position:**

Yes

**Position In Title:**

Yes

**Related Work:**

4

**Strengths And Weaknesses:**

## Strengths

1. The paper is well written, and the position is grounded in evidence, and is important especially in high-stakes tasks.
2. The paper presents a thorough examination of existing literature and contrasts the proposed KO mechanism with other existing approaches in AI systems.

## Weaknesses

My main issue is with the KO mechanism itself, though it might stem from me misunderstanding certain aspects. While I buy the general position the paper is proposing, the KO mechanism feels like it will simply add to the amount of information needed for retrieval when processing a task. More concretely:
1. The connection between the KO and the established LLM weaknesses (hallucinations, miscalibrations, prompt sensitivity) is still unclear. KOs seem to represent associations LLMs have made implicitly from the external knowledge, rather than the reasoning mechanisms. So the effectiveness of the KO really depends on the granularity and the weight of the evidence.
For instance: an LLM may give a very hand-wavy set of evidence to support a claim it is making, but that does not mean it actually reasoned about the evidence.
Alice may evaluate whether the claim is correct wrt the evidence, but again, Alice is evaluating pattern correlations, not reliability of reasoning.

2. How are KOs different from other AI generated objects that try to explain the reasoning behind their outputs? e.g. CoTs And how is this different from an agentic memory system with a human in the loop, where before something is stored in memory a human validates it?
Structure seems to be the only discriminating feature of KOs in that sense.

3. There is still an implicit gap when it comes to reasoning on new tasks: the LLM may not realize certain KOs are relevant to a task, simply choose to ignore the KO,
or still make reasoning mistakes by inappropriately applying the KO.
Furthermore, since it is well established that LLM performance drops with increase in context length, if the number of KOs relevant to a task is very high, won't that potentially reduce the LLM's capabilities?

3. One significant bottleneck not discussed is the scale of the KOs at an organizational level. Depending on the granularity of the KO, one user's request might trigger the production of
thousands of KOs. So in subsequent responses, the LLM needs to not only retrieve the relevant information to the user's query, but also figure out which KOs it needs to adhere to
and retrieve those. It seems like the data cost will very quickly blow up.

**Support:**

3

---

> ### Author Rebuttal · Authors · 2026-03-31
>
> We thank the reviewer for the constructive comments.
>
> To clarify the scope of KOs, we decompose AI system performance as follow:
>
> > **AI System Task Outcome Quality = LLM Capability × Context Relevance × Context Quality**
>
> These three dimensions are largely orthogonal [1,2,3]: LLM Capability is model-intrinsic, Context Quality is task-independent, and Context Relevance is task-dependent. (See the response to Reviewer mZZH Q1 for a theoretical grounding.) **Validated KOs tend to be high-quality context, thereby effectively improving AI system task outcome quality.**
>
> **W1.** LLM weaknesses (hallucination, miscalibration, prompt sensitivity) are *symptoms* with multiple sources across all three dimensions. Context Quality plays a critical role: noisy or incorrect context can cause hallucination; ambiguous or contradictory context can cause a model to overgeneralize, producing miscalibrated confidence; poorly organized context can cause a model extract different information under minor query variations, amplifying prompt sensitivity. **KO mitigates these symptoms by improving Context Quality**: "Verifiable" ensures context undergoes human validation, catching inaccuracies; "Understandable" reduces ambiguity via structured expression; "Controllable" lets humans correct contradictions; "Traceable" reduces surface-form dependence and enables source tracing when context is incomplete. Recent work also confirms the role of Context Quality: CL-bench [1] shows that knowledge organization affects model performance significantly — more than a 25% gap between task categories within the same domain; GSM-Symbolic [4] shows misleading context degrades performance by up to 65%.
>
> The reviewer's observation that KO effectiveness depends on granularity and evidence is valid. We note that KOs are not externalized from an LLM's implicit knowledge of external documents. As stated in Section 4.3: "From this data, the AI System externalizes implicit knowledge into structured Knowledge Objects" — the input is **Human-AI interaction data**, not external knowledge. This distinction is critical: interaction data is produced through human participation, so humans possess the **ground truth** to judge KO quality and granularity.
>
> Therefore, Alice's validation is not "evaluating pattern correlations" between LLM evidence and LLM claims. Alice validates based on **her own experience from the collaboration** — she encountered the singleton thread-safety issue in her production environment. If evidence is hand-wavy, Alice can supplement it, or she can reject the KO — the validation status is explicitly recorded.
>
> **W2.** We would like to clarify that KOs do not explain the reasoning behind LLM outputs, which is done by CoTs. KOs capture validated knowledge externalized from Human-AI interaction data — persistent and reusable across sessions, unlike CoTs which is ephemeral and tied to a single generation.
>
> On Agent Memory + Human-In-The-Loop (HITL), in the three-factor framework, Agent Memory and KOs both belong to the context but are positioned differently. Agent Memory is a context **type** (alongside skills, knowledge graphs, etc.). KOs define **properties** that context must satisfy for Reliable AI. Any context satisfying these five properties qualifies as a KO — regardless of its type. Agent Memory + HITL is therefore a **means** to implement KOs, not KOs by themselves.
>
> More importantly, as discussed in Section 5.2, the question is not whether validation *could* be added to agent memory, but whether such additions would produce reliable systems. Agent memory asks "How do we store information so the AI can find it?"
> Human validation is not a first-class concern in this design. In contrast, KOs treat Reliable AI as a first-class design constraint: “How can context be made trustworthy and verifiable for humans?” These two starting points lead to fundamentally different systems.
>
> Our position is that the community should explore this latter direction — **making Reliable AI properties first-class design constraints rather than afterthoughts**. This is a core value of a position paper.
>
> **W3 and W4.** W3 (LLM ignoring/misapplying KOs, long-context degradation) and W4 (scalability/retrieval cost) primarily fall under **LLM Capability** and **Context Relevance**. Both are important challenges, but our paper focuses on a different dimension, what properties make context trustworthy (Context Quality). How these properties are implemented, retrieved, and scaled across specific context types is an important direction that we leave for future research.
>
> **References:**
>
> [1] CL-bench: A benchmark for context learning. *arXiv* 2026.
>
> [2] FRAMES: Fact, Fetch, and Reason. *NAACL* 2025.
>
> [3] ACE: Agentic context engineering. *ICLR* 2026.
>
> [4] GSM-Symbolic: Understanding the limitations of mathematical reasoning in LLMs. *arXiv* 2024.

---

> > ### Author Rebuttal · Reviewer_Qz8u · 2026-04-02
> >
> > I thank the authors for addressing my concerns.
> >
> > Just a couple of clarifications: When I say associations LLMs have made implicitly from the external knowledge, I mean the associations implicit in the data the LLM was trained over, not that additional knowledge is provided as an additional input. Regardless, I think your clarifications still apply, and I greatly appreciate them.
> >
> > Your response makes it sound like KOs are explicitly for context quality dimension of general AI reliability. This turns the position into a much narrower one than KOs being central for general AI system reliability; the other dimensions are equally important, and introducing KOs in its idealized form does not still resolve reliability without significant research in the other dimensions. I want the paper to make this distinction clear, specifying that this is one of three dimensions contributing to the reliability of AI systems (either that, or present an argument as to why you aren't overclaiming the role of KOs).
> >
> > I also think W3 and W4 still deserve a discussion, and I would like you to elaborate on it. While it may be orthogonal to context quality, the reliability of the AI system is still heavily dependent on improvements in these other dimensions. Such a discussion should include how KOs can shape research along these dimensions.
> >
> > Overall, I like the paper as a position paper, even if I am skeptical of KOs in general. But you must either provide a sufficient justification for the current claims, or describe how you would readjust the scope of the position, and discuss its impact on the other dimensions.

---

### Official Review · Reviewer_mZZH · 2026-03-18

**Significance:** 4
**Argument Clarity:** 4
**Rating:** 5
**Confidence:** 4

**Questions:**

- As AI models are quickly updated (e.g., new pretraining data and different post-training recipes), how are we supposed to stabilize the design and implementation of Knowledge Objects.

**Alternative Views Section:**

Yes

**Compliance With Llm Reviewing Policy A Conservative:**

Affirmed.

**Discussion Potential:**

3

**Paper Summary:**

This paper proposes a position to externalize implicit model knowledge via 'knowledge objects' (KOs), so that humans can inspect, verify, and endorse. As opposed to classic methods like RAG, the proposed exploration centers on AI-generated knowledge, structured for human inspection. Alternative approaches like self-verification and agent memory also suffer from various deficiencies: knowledge is unexposed manifesting with only confidence scores (self-verification) and raw/unstructured data with no validation status (agent memory).

**Position:**

Yes

**Position In Title:**

Yes

**Related Work:**

3

**Strengths And Weaknesses:**

Pros
- Well-structured, easy to follow, and I especially appreciate the clear comparison with existing solutions (e.g., RAG and agent memory).
- How to steer and trust AI systems is a profound question, and the authors propose a viable way to bring human-in-the-loop.
- Alternative views (e.g., 'Human Validation Will Bottleneck AI Productivity') are well constructed and well rebutted.

Cons
- It would be even better if the authors could provide benchmarks or sketch milestones so that the community's efforts towards externalizing implicit knowledge can be meaningfully measured and tracked.
- Although I very much agree on the importance of externalizing implicit model knowledge, I am not quite sure if the proposed Knowledge Objects are general and extensible enough to be widely implemented. For example, systematically checking the scope and limitations is highly non-trivial to me. It is not clear how humans are supposed to handle those limitations when examining the claims in Knowledge Objects.

**Support:**

3

---

> ### Author Rebuttal · Authors · 2026-03-31
>
> We thank the reviewer for the positive evaluation and constructive suggestions.
>
> **W1.** Existing studies exhibit fragments of a governance-maturity spectrum — from raw storage [1,2] through genuine extraction [3,4] — but none provides a unified governance target for Reliable AI. **KO's five properties define the missing target** — when any information satisfies all five, it becomes trustworthy explicit knowledge, providing an anchor for leveraging implicit knowledge and achieving Reliable AI. Based on this, we sketch the following milestones in Section 6:
>
> - *Near-term*: **Single-domain KO prototypes** — build on existing extraction systems (ExpeL [3], Voyager [4]) as the extraction backbone, and add the five KO properties on top: validation metadata, provenance tracking, human review interface. Evaluate whether validated KOs reduce downstream error rates compared to unvalidated context.
> - *Mid-term*: **Cross-domain KO framework** — abstract from single-domain experience to define a domain-agnostic KO schema, standardize how five properties are represented across domains, and establish cross-domain evaluation protocols.
> - *Long-term*: **Community-wide KO ecosystem** — shared KO repositories (analogous to model hubs but for validated knowledge), community validation workflows (analogous to Wikipedia's review process), and cross-system interoperability enabling KOs from one system to be reused in another.
>
> **Q1.** To explain why KOs remain stable under model updates, we decompose AI system performance as three dimensions:
>
> > **AI System Task Outcome Quality = LLM Capability × Context Relevance × Context Quality**
>
> The multiplicative form reflects that the weakest factor dominates overall quality. LLM Capability captures the model's intrinsic reasoning and generation abilities. Context Relevance captures task-dependent properties such as pertinence, sufficiency, and timeliness [10]. And Context Quality comprises two sub-dimensions: *performance-oriented properties* (accuracy, consistency, format, structure, signal-to-noise ratio) that primarily serve task performance [7,8,9], and *Reliable AI properties* (verification status, traceability, controllability, reusability, understandability) that primarily serve reliability — **KO's five properties correspond precisely to the Reliable AI sub-dimension.** Satisfying KO properties requires human participation, which naturally improves performance-oriented properties as well.
>
>
> Recent work [5,6,11] supports the orthogonality of these three dimensions. Thus model updates (which change Capability) do not affect validated KOs (which belong to Context Quality). E.g., "Singleton pattern causes thread-safety issues under high concurrency" remains useful whether applied to GPT-4, GPT-5, or future GPT-6.
>
> **W2.** KO's five properties belong to the Reliable AI dimension of Context Quality — they define *properties* context must satisfy, not a context *type*. Context types will proliferate as technology evolves, but the core requirements for trustworthy context will not — just as safety standards do not become obsolete when new product categories emerge.
>
> On the non-triviality of checking scope and limitations, KOs are extracted from Human-AI interaction data — humans participated in the collaboration that generated this data. Therefore, humans possess the **ground truth** to assess scope and limitations, because the knowledge originates from their own practice. Moreover, this checking process is itself a form of **human learning** — validating KOs is not an extra burden but part of knowledge internalization for humans.
>
> For example, when Alice collaborates with an AI on a code review and the system externalizes a KO ("singleton causes thread-safety issues under high concurrency; scope: Java/Spring, >100 req/s"), Alice can assess the scope's accuracy and add limitations (e.g., "does not apply to read-only replicas") based on her experience — because she accumulated that experience through collaboration that produced the KO.
>
> **References:**
>
> [1] Reflexion: Language agents with verbal reinforcement learning. *NeurIPS* 2023.
>
> [2] MemGPT: Towards LLMs as operating systems. *arXiv* 2023.
>
> [3] ExpeL: LLM agents are experiential learners. *AAAI* 2024.
>
> [4] Voyager: An open-ended embodied agent with LLMs. *Trans. Mach. Learn. Res.* 2024.
>
> [5] FRAMES: Fact, Fetch, and Reason. *NAACL* 2025.
>
> [6] ACE: Agentic context engineering. *ICLR* 2026.
>
> [7] GSM-Symbolic: Understanding the limitations of mathematical reasoning in LLMs. *arXiv* 2024.
>
> [8] Lost in the middle: How language models use long contexts. *TACL* 2024.
>
> [9] Enhancing noise robustness of retrieval-augmented language models. *ACL* 2024.
>
> [10] Sufficient context: A new lens on RAG systems. *ICLR* 2025.
>
> [11] CL-bench: A benchmark for context learning. *arXiv* 2026.

---

> > ### Author Rebuttal · Reviewer_mZZH · 2026-04-02
> >
> > Thank you for the elaborated answers. All my concerns have been resolved. :-)

---

### Decision · Program_Chairs · 2026-04-30

**Decision:**

Accept (regular)

**Comment:**

The reviewers agree that the authors have mostly answered their criticisms successfully.